# Microhabitat and ectomycorrhizal effects on the establishment, growth and survival of *Quercus ilex* L. seedlings under drought

**Laura García de Jalón**[1]*, **Jean-Marc Limousin**[1], **Franck Richard**[1], **Arthur Gessler**[2], **Martina Peter**[2], **Stephan Hättenschwiler**[1], **Alexandru Milcu**[1,3]

**1** CEFE, Univ. Montpellier, CNRS, EPHE, IRD, Univ. Paul Valéry Montpellier 3, Montpellier, France, **2** Swiss Federal Research Institute WSL, Birmensdorf, Switzerland, **3** Ecotron Européen de Montpellier (UPS-3248), CNRS, Montferrier-sur-Lez, France

* laura.garciadejalon@gmail.com

**Data Availability Statement:** All relevant data are within the manuscript and its Supporting Information files.

## Abstract

The success of tree recruitment in Mediterranean holm oak (*Quercus ilex*) forests is threatened by the increasing intensity, duration and frequency of drought periods. Seedling germination and growth are modulated by complex interactions between abiotic (microhabitat conditions) and biotic factors (mycorrhiza association) that may mitigate the impacts of climate change on tree recruitment. To better understand and anticipate these effects, we conducted a germination experiment in a long-term precipitation reduction (PR) field experiment where we monitored seedling establishment and survival, micro-habitat conditions and ectomycorrhizal (ECM) colonization by different mycelia exploration types during the first year of seedling growth. We hypothesized that (i) the PR treatment decreases seedling survival relative to the control with ambient conditions, (ii) microhabitat conditions of water and light availability are better predictors of seedling survival than the PR treatment, (iii) the PR treatment will favour the development of ECM exploration types with drought-resistance traits such as differentiated rhizomorphs. Contrary to our first hypothesis, seedling survival was lower in control plots with overall higher soil moisture. Micro-habitat light and soil moisture conditions were better predictors of seedling survival and growth than the plot-level PR treatment, confirming our second hypothesis. Furthermore, in line with our third hypothesis, we found that ECM with longer extramatrical mycelia were more abundant in the PR treatment plots and were positively correlated to survival, which suggests a potential role of this ECM exploration type in seedling survival and recruitment. Although summer drought was the main cause of seedling mortality, our study indicates that drier conditions in spring can increase seedling survival, presumably through a synergistic effect of drought adapted ECM species and less favourable conditions for root pathogens.

## Introduction

The success of seedling establishment is one of the most critical processes determining forest regeneration dynamics and long-term persistence of tree species. In Mediterranean forests,

**Funding:** AM received ANR (Agence Nationale de la Recherche, France) - funded TRANSfER project (ANR-16-CE32-0002). https://anr.fr/en/ The funders had no role in study design, data collection and analysis, decision to publish, or preparation of the manuscript.

**Competing interests:** The authors have declared that no competing interests exist.

where summer drought is already a major cause of mortality of woody species during early life stages [1, 2], the success of tree establishment is expected to deteriorate as a consequence of the ongoing climate change. Climate models for the Mediterranean region predict longer and more severe drought episodes in the near future [3] and the first signs of climate change impacts on forests are already documented as tree dieback and crown defoliation [4, 5]. As Mediterranean forests are, for the most part, naturally regenerated, a better understanding and quantification of drought effects on tree recruitment processes is needed to better mitigate the impacts of climate change on forest ecosystems.

In addition to the climatic effects, the success of tree recruitment under a forest cover is also highly dependent on microhabitat conditions [6–8] and biotic interactions, especially with mycorrhizal fungi [9, 10]. The canopy cover and structure generate different microhabitats, for example through different light availabilities that further modulate local soil moisture and temperature conditions, and thus, the overall effect of drought. The canopy can act as shelter in the dry season by protecting against radiative heating, reducing soil temperature, and evaporative demands [11]. Conversely, the tree canopy can also reduce the soil moisture available to seedlings due to foliar interception of rainfall, and an increased competition for soil water by the adult trees [12, 13]. In addition to these subtle differences that tree canopies can cause on water availability of understorey seedlings, canopy cover also has a direct effect on light availability. Two studies carried out with *Quercus pyrenaica* Wild. in different forest ecosystems of Spain with different water availabilities revealed higher seedling survival under open canopies in the wet sites [14], and under closed canopies in the dry sites [15]. This suggests a trade-off between light availability and soil water status, which is in line with the morphological trade-offs between plant tolerance to drought and shade [16]. However some species, such as *Quercus ilex*, are known to tolerate both shade and drought, with the shade contributing to expand their drought tolerance [17]. These species might be favoured in Mediterranean ecosystems where light and water limitations co-occur, especially in the case of seedlings emerging under a dense tree canopy. In theory, the most suitable microhabitat for seedling establishment and growth is a site with enough light to maintain a positive carbon balance, and with sufficient soil moisture to avoid water deficit during the drought seasons. However, the impact of water availability on seedling recruitment is complex and not yet fully understood, with opposite effects of rainfall exclusion being documented for seedling emergence (increased in drier conditions) and seedling survival (decreased in drier conditions) [18].

There is also increasing evidence that the filamentous fungi known to form plant-fungal chimeras, the so-called mycorrhizae, are critical not only for nutrient uptake and defence against soil pathogens but also for helping plants to cope with the detrimental effects of soil water deficit [9, 10, 19]. Sheltering adult trees may also facilitate seedling establishment by acting as a fungal inoculum for seedlings and providing a common mycelium network which can be beneficial for the performance of younger generations of plants [20–22]. The widespread Mediterranean tree species *Q. ilex* is associated with ectomycorrhizal fungi (ECM). In forest ecosystems, these ECM fungi are assembled in highly diverse communities [23] made of patchily distributed species [24]. Species composition of these communities are influenced by water availability, and have been shown to respond to precipitation reduction [25]. Depending on fungal species, ECM vary in the anatomy and morphology of the mycelia radiating into the soil and these morphological variations have been putatively related to ecological functions and types of foraging exploration [26]. According to [26], four different exploration types can be distinguished based on the amount of emanating hyphae: contact, short, medium-distance and long-distance. It has been suggested that long exploration-type ECM root tips with hydrophobic mantles and highly differentiated extensive external mycelium (i.e., the so-called

rhizomorphs, developed by species in *Boletus* and *Cortinarius* genera), are resistant to drought and effective at water transport and uptake for their associated hosts [27, 28]. However, to the best of our knowledge, the potential interactive effects of microhabitat environmental factors with ECM colonization and exploration types on seed germination and seedling establishment have been little investigated, especially under field conditions. Most studies on seedling ECM colonization have been done in highly artificial conditions (potted plants in greenhouses, applied fungal inoculums, simplified soil substrates) which may be prone to being affected by lab-specific environmental factors [29] and thus, their relevance for natural field conditions should be interpreted with caution.

In this study, we investigated the effects of increasing drought conditions on the germination, growth and survival of *Q. ilex* seedlings in a long-term field experiment with precipitation reduction (PR) of 29% compared to the unaltered control since 2003. We conducted a germination experiment in the field and monitored seedling establishment and survival, soil humidity and light micro-conditions and ECM colonization of roots during the first year of seedling growth. We hypothesized that: (i) the PR treatment decreases seedling survival relative to the control with ambient conditions [18], (ii) microhabitat conditions of water and light availability are better predictors of seedling survival than the PR treatment, (iii) the PR treatment will favour the development of ECM fungi with drought-resistance traits such as differentiated rhizomorphs with longer external mycelium.

## Materials and methods

### 1) Experimental site

The study site is located 35 km northwest of Montpellier (southern France) in the Puéchabon State Forest, on a flat plateau (43˚44'29" N, 3˚35'45" E, elevation 270m). This forest has been managed as a coppice for centuries and the last clear cut was performed in 1942. Vegetation is largely dominated by the shade-tolerant evergreen oak *Q. ilex*, with a top canopy height of about 5.5 m and a stand density of *c*. 6000 stems ha$^{-1}$. The understorey is composed of a sparse shrubby layer with the evergreen species, *Buxus sempervirens*, *Phyllirea latifolia*, *Pistacia terebinthus* and *Juniperus oxycedrus*, a percent cover of approximately 25% at 2 m.

The area has a Mediterranean-type climate. Rainfall mainly occurs during autumn and winter with about 80% of total annual precipitation occurring between September and April. The mean annual precipitation is 920 mm with a range of 550–1549 mm (1984–2017). Mean annual temperature is 13.3˚C (on-site meteorological station, 1984–2017), the coldest month being January (5.5˚C) and the hottest July (22.9˚C). The soil is extremely rocky from hard Jurassic limestone origin. The average volumetric fractional content of stones and rocks is about 75% for the top 0–50 cm and 90% below. The stone free fine fraction within the top 0–50 cm layer of the soil is a homogeneous silty clay loam (USDA texture triangle, 38.8% clay, 35.2% silt and 26% sand). Seed germination in *Q. ilex* mostly occurs in early winter and shoot emergence in spring. The below-ground establishment occurs from germination to shoot emergence, when plants invest in their taproot and when the first associations between roots and below-ground microbial communities (ECMs and/or pathogens) take place.

### 2) Rainfall exclusion

In March 2003, a partial throughfall exclusion experiment was set up on the site. The throughfall exclusion experiment was replicated on three blocks 200 m away one from the other, and situated on a flat area with no lateral runoff. Each replication was composed of one throughfall exclusion treatment (henceforth, precipitation reduction, PR) and one control treatment (henceforth, Control), each with a plot area of 140 m$^2$ (14 m x 10 m). Throughfall exclusion is

achieved by using 14 m long and 0.19 m wide PVC gutters covering 33% of the ground area underneath the tree canopy. Taking into account interception losses by the canopy and stem-flow, the throughfall exclusion treatment effectively reduces the net input of precipitation to the soil by 29% compared with the control treatment [30]. On the control plots, identical gutters are set up upside down so that the albedo and the understorey microclimate are as close as possible in the two treatments. However, gutters in the PR treatment are slightly tilted down in order to let the water drop out of the plots, different from the Controls where gutters are horizontally placed.

### 3) Sowing, establishment and seedling growth

Acorns from 10 different maternal trees were collected in November 2017 in the surroundings of the experimental site, outside of the precipitation reduction and control treatments. All acorns were submitted to a flotation test in order to discard the ones aborted or damaged by insects or fungi. All the viable acorns were weighted, numbered with a marker pen for identification, and kept in a fridge at 4˚C in vermiculite substrate humidified with water for 3 weeks. Sowing was done in December 2017, placing acorns 1 cm deep into the soil in the understorey of the experimental site. In order to prevent acorn predation, sowed acorns were enclosed in wire cages of 40 × 30 × 30 cm. One acorn from each maternal tree was sowed in each cage, thus totalling 10 acorns par cage. A total of 60 cages were randomly installed in areas not occupied by stones (30 cages per treatment, 20 per block), so 600 acorns in total were used in the study.

Recruitment stages (germination, stem emergence and survival) were monitored monthly from January 2018 to the end of November 2018 in all seedlings from all cages. Two metrics of survival, and hence mortality, were considered in the study: survival from germination to the collection of the seedlings after 11 months was considered as 'total survival' ($S_T$), and survival from stem emergence to the collection of seedlings after 11 months was considered as 'survival of emerged plants' ($S_{EP}$). Stem height was measured in all seedlings from all cages at the end of the experiment in November 2018. At this time, all seedlings from half of the cages randomly chosen (15 cages per treatment, 10 per block) were pulled out from the soil for ECM analysis and biomass assessments. Aerial and belowground seedling parts were separated, the aerial parts were oven-dried at 60˚ C until constant mass for biomass measurements, and belowground parts were used for ECM assessments.

### 4) Microenvironmental conditions

Soil water content and light availability were measured six and two times respectively during the one-year experiment for each of the cages/microhabitats. Two measurements per cage were taken at two different positions of the cage that were averaged to get the final value for both soil water content and light measurements at each measuring time. Volumetric soil water content (SWC) was measured six times in the year in the first 10 cm of soil with a portable TDR (Delta-T SM150 Soil Moisture kit, Cambridge, UK), and light availability was measured in Spring and Autumn with a LAI-2200 Plant Canopy Analyser (LI-COR Biosciences, Lincoln, NE, USA). As the plant or leaf area index over the cages could not be measured because of the shading from the gutters installed in the plots, the canopy gap-fraction (GF) around a zenith angle of 38˚ was used as an integrative proxy of the local light availability. Simultaneous measurements of incident light above and below the canopy where taken with the LAI-2200, and the GF was computed using the FV2200 2.1.1 software (LI-COR Biosciences). GF was preferred to a simple ratio between the light below and above the canopy so as to avoid the

confounding effects due to differences of the solar angle or of the fraction of direct and diffuse light at the time of measurement.

## 5) ECM quantification

At the end of November 2018, eleven-month old seedlings (alive and dead) from half of the cages of both treatments and blocks (30 cages and 129 plants analysed) were carefully dug up by hand with the aim to collect the largest part of the root systems down to 15 cm depth. Due to the very rocky nature of the soil, it was not possible to harvest roots that were growing deeper than 15 cm. As it was not possible to extract intact root systems, the root biomass variable was omitted from the ECM analysis. Seedlings were brought to the lab and kept in the fridge for a week before ECM quantification under a dissecting microscope. The collected root system was kept in water in trays for several hours in order to remove organic and mineral soil particles from the roots. On the extracted root systems, the percentage of root colonization by ectomycorrhizal fungi was quantified under the microscope, by counting the number of colonized and non-colonized root tips. As the assessment of the ECM colonization was feasible irrespective of the plant status (alive or dead), we quantified it in both alive and dead seedlings. For dead seedlings this measurement corresponds to the status of ECM colonization at the moment of their death and it constitutes a valuable information for assessing the role of ECM in seedling survival since most *Q. ilex* seedlings die during their first year, particularly under adult trees [7, 22]. Each ECM root tip was classified according to the different exploration types defined by [26] based on the amount, organization and length of emanating hyphae (contact, short, and medium-distance). Proportions of each exploration type were calculated by dividing the number of ectomycorrhizae of each exploration type respective by the total number of root tips of the root system (Eqs 1 and 2).

$$ECMc = \frac{n° \text{ contact root tips}}{n° \text{ total of root tips}} \tag{1}$$

$$ECMsm = \frac{n° \text{ short } + \text{ medium root tips}}{n° \text{ total of root tips}} \tag{2}$$

With ECMc being the proportion of contact exploration type root tips and ECMsm the proportion of short and medium exploration type root tips. In line with the study of [27] that was carried out at the same experimental site, we did not find any long-distance ECM associated to the root tips of *Q. ilex* seedlings, thus explaining the absence of this exploration type in our dataset.

## 6) Statistical analyses

All statistical analyses were performed with the R software version R 3.4.1 (R Core Team, 2017). Acorn germination, survival and growth-related response variables were analysed with generalized linear mixed-effects models with the PR treatment as a fixed factor and soil water content (SWC), canopy gap fraction (GF) and acorn mass as covariables, with the random effects being the block, the cage and the mother tree using *glmer* function from 'lme4' package version 1.1–21 [31]. The SWC and GF covariables were measured at the cage level, whereas the acorn mass covariable was measured at the individual level. An example of the syntax for a model looking at germination success is as follows "*model<- glmer (Germination ~ PR_treat + SWC * GF * acorn_mass + (1|Block) + (1|Cage) + (1|ID_mother_tree), family = binomial)*". Binomial variables such as germination and survival were modelled with binomial distribution, and proportion data such as ECM colonization with betabinomial distribution. The

seedling status at the time of harvest (dead or alive) was also added as an additional explanatory variable in models of ECM colonization. Minimal adequate models with the lowest AIC were obtained following the guidelines of [32] with the help of the "buildmer" package version 1.4 [33]. Interaction between ECM colonization and environmental variables were displayed with the "interactions" package version 1.1.1 [34], and the survival curves with the "survival" package version 3.1–8 [35]. The overall effects of the PR treatment on SWC and GF across all sampling campaigns were analysed using linear mixed-effects models with PR treatment and measurement campaign (sampling date) as fixed factors and cage as random effects, whereas for the models testing the effect of PR treatment for each sampling campaign/date only the block was used as random effect.

## Results

### 1) Effects of precipitation reduction treatment on microhabitat conditions

Volumetric soil water content measured in the upper soil layer (10 cm) at the microhabitat/cage level were significantly lower (-11%) in the precipitation reduction treatment (PR) relative to Control over all six measurement campaigns (P<0.002; Fig 1). The same trend was found also in all individual measurement campaigns (Fig 1). The lowest soil water content values were observed in August and in September (2.1% in Control and 1.2% in PR in September) during the peak of the seasonal drought, thus confirming that 2018 had a typical summer drought for the Mediterranean region. The highest soil water content values were observed in April, with 27% in Control and 26% in PR treatment. There was a large variability in SWC among cages with a strong overlap of values between treatments, irrespective of the measurement period.

Similar to SWC, we observed a large variability and an important overlap of cage level GF values between treatments, indicating highly different light conditions among microhabitats

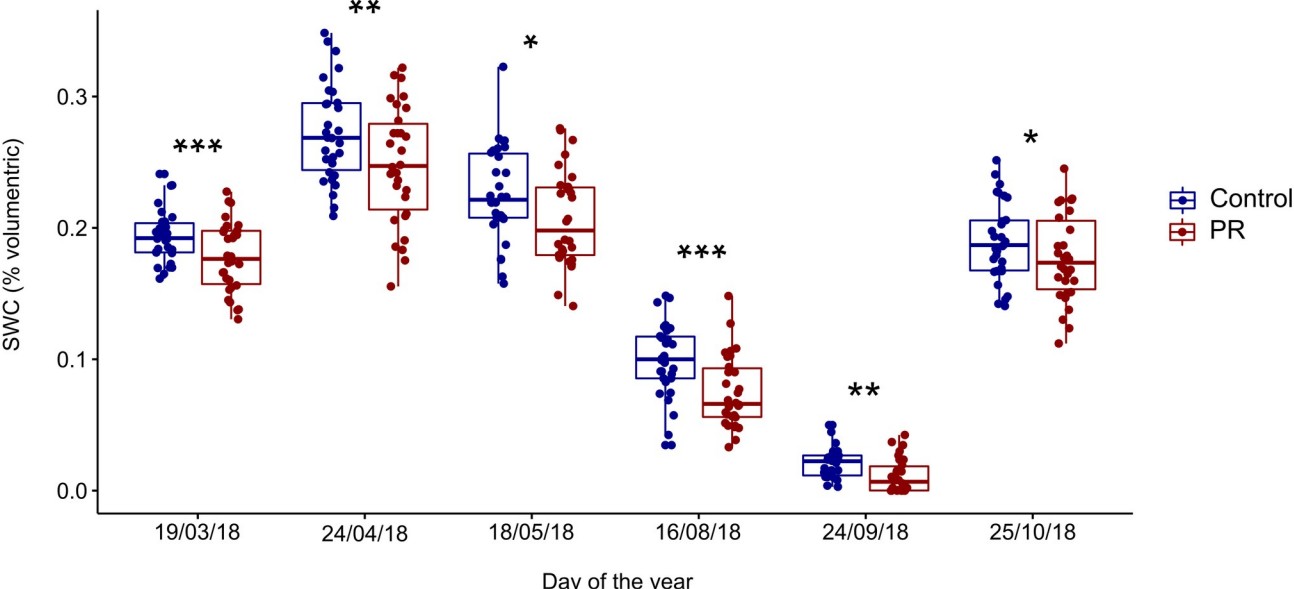

**Fig 1. Volumetric soil water content (SWC % volumetric) per treatment measured from March to October 2018.** Precipitation reduction treatment (PR, in red) and control (in blue). Each dot indicates the average SWC value in one of the 60 cages. *** for P<0.001, ** for P<0.01, * for P<0.05, + for P<0.07.

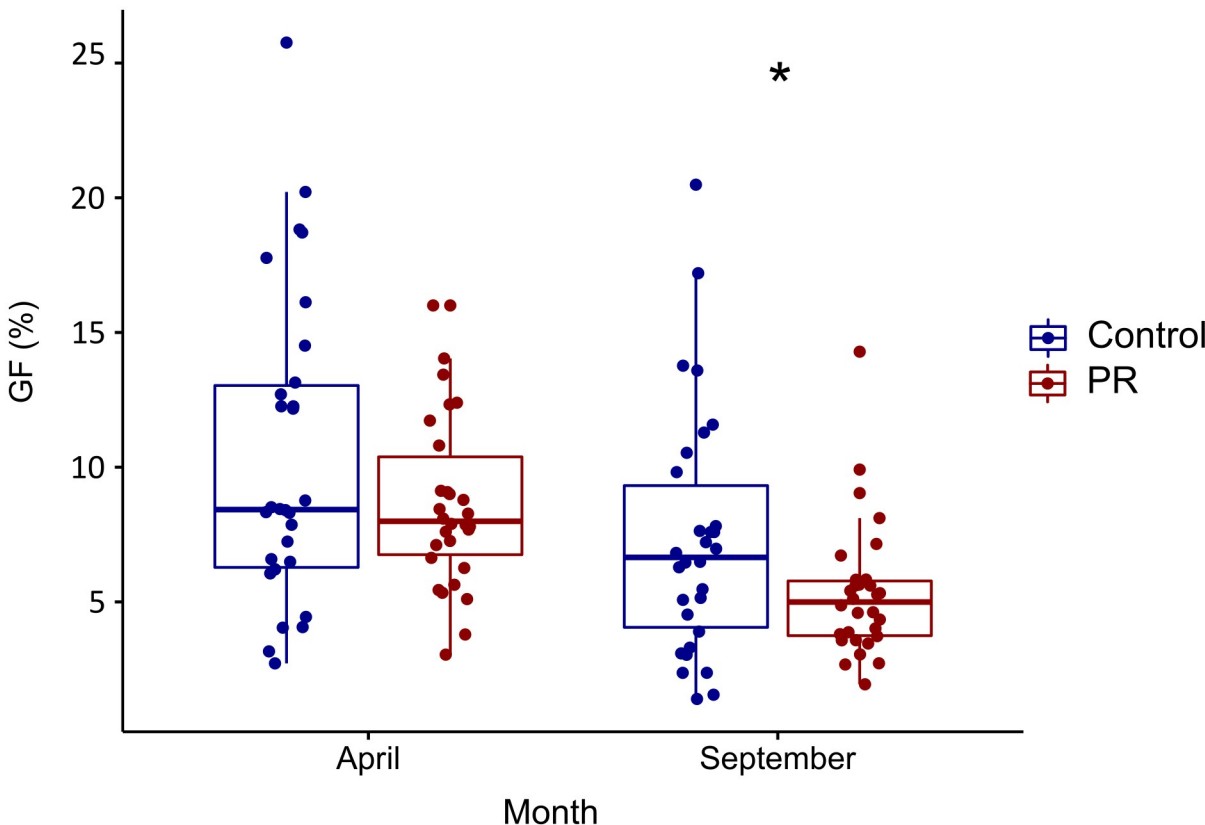

**Fig 2. Gap fraction (%) per treatment measured in April and September 2018.** Precipitation reduction treatment (PR, in red) and control (in blue). Each dot indicates one microhabitat at each of the 60 cages. *** for P<0.001, ** for P<0.01, * for P<0.05, + for P<0.07.

(Fig 2). Over the two sampling campaigns, the PR treatment, in which gutters are slightly tilted down and therefore closer to cages, showed an overall lower but not significant GF value compared to the Control (7% and 8.8%, respectively) (Fig 2). However, the effect of PR treatment on GF was not significant in April when *Q. ilex* only supported its old leaf cohorts, but was significantly lower in September (Fig 2).

## 2) Germination and survival

The proportion of germinated acorns increased with the initial seed resources (acorn fresh biomass) (Table 1) but was not significantly affected by the precipitation reduction treatment (PR), nor by any of the measured micro-environmental variables.

Total survival ($S_T$) of seedlings after eleven months was significantly higher (29%) in the PR treatment relative to Control (22%) (Figs 3 and 4B), and was higher in seedlings from bigger acorns and growing under more open canopies (Table 1). The survival of emerged plants ($S_{EP}$) (i.e. the survival of seedlings that already succeeded germination and stem emergence) also increased with seed resources (acorn biomass), and it was significantly lower in the wetter microhabitats (Table 1; S1 Fig), with no influence of light. Seedling mortality during germination was similar in seedlings from different drought treatments (25.7% Control vs. 23.7% in PR, respectively). Mortality in spring when below-ground establishment and stem emergence occurred was slightly higher in the Control than in PR treatment (15.6% Control vs. 12% PR

**Table 1. Fitted estimates and significance levels of linear mixed models for seedling germination, survival and growth.** Minimal adequate model outputs for the effects of precipitation reduction (PR) treatment, acorn fresh biomass (Acorn), gap fraction (GF), soil water content (SWC) and their respective interactions on germination, total survival ($S_T$), survival of emerged plants ($S_{EP}$), height, shoot biomass, root biomass of the extracted root systems, contact ectomycorrhizal colonization (ECMc) and short + medium ectomycorrhizal colonization (ECMsm). $R^2m$ represents the variance explained by fixed factors and $R^2c$ represents the variance explained by fixed and random factors. NA = non applicable, as $R^2c$ cannot be computed for beta binomial distributions. *** for $P<0.001$, ** for $P<0.01$, * for $P<0.05$, + for $P<0.07$. Blanks indicate that the respective variables was not retained in the minimal adequate models, whereas the estimates with a lack of symbol (star or plus) are not statistically significant, but the predictor was retained in the minimal adequate model.

| Variable/ Predictor | PR | Acorn | GF | SWC | Acorn×GF | GF×SWC | $R^2$m | $R^2$c |
|---|---|---|---|---|---|---|---|---|
| Germination | | 0.231+ | 0.1672 | | | | 0.02 | 0.29 |
| Total survival ($S_T$) | 0.482* | 0.272** | 0.211* | | | | 0.04 | 0.11 |
| Survival emerged plants ($S_{EP}$) | | 0.238* | | -0.203* | | | 0.03 | 0.03 |
| Height | -1.468* | 1.233*** | | | | | 0.15 | 0.27 |
| Shoot biomass | | 0.042*** | | -0.016* | | | 0.22 | 0.22 |
| Root biomass | | 0.047*** | | | | | 0.23 | 0.23 |
| ECM$_c$ | | | -0.231* | 0.148+ | | 0.496*** | 0.15 | NA |
| ECM$_{sm}$ | 0.536*** | | -0.192* | | -0.263** | | 0.12 | NA |

respectively, difference not significant). The highest seedling mortality was observed during the summer drought season and was again slightly higher in the Control than in the PR treatment (33.7% Control vs. 31.3% PR respectively, difference not significant). Finally, mortality in autumn during the drought recovery was modest in comparison and similar between treatments (3.3% in Control vs. 3.7% in PR treatment).

### 3) Growth

The height of the surviving seedlings was lower (-10%) in the PR treatment compared to Control (Table 1; Fig 4C). Heavier acorns with more initial seed resources were observed to produce taller and heavier seedlings in both treatments (Table 1). Shoot biomass was negatively influenced by the soil moisture of the micro-sites, but it was not significantly impacted by the PR treatment (Table 1; Fig 4D). Root biomass of the extracted root systems was positively influenced by acorn biomass (Table 1). Contrary to soil humidity, light availability at the microsite level did not have any influence on seedling growth (Table 1).

### 4) Ectomycorrhizal colonization and correlations with seedling performance

Ectomycorrhizal colonization assessed as the proportion of mycorrhizal root tips per seedling ranged from 61 to 100%. Overall, the proportion of contact exploration type (ECMc) was slightly but not significantly lower in the PR treatment (-6.5%; Fig 5A) relative to the Control, whereas the proportion of short and medium exploration types (ECMsm) was significantly larger in the PR treatment compared to the Control (+59.7%; Fig 5B). The status of seedlings (dead or alive) as covariable in the models was not significant for any of the ECM exploration types. In terms of micro-environmental conditions, the colonization with ECMc was affected by the interaction between soil moisture and light availability (Table 1; Fig 6A); the proportion of ECMc increased under more optimal conditions with high SWC and light availability, but decreased with light availability when SWC was low. The colonization of root tips with ECMsm was affected by the interaction between acorn biomass and light availability (Table 1; Fig 6B). ECMsm was high for seedlings with low seed biomass, with a slight increase of proportions of ECMsm with increasing light availability. In contrast, in seedlings originating from heavy acorns, ECMsm decreased with increasing light availability. In other words, seedlings

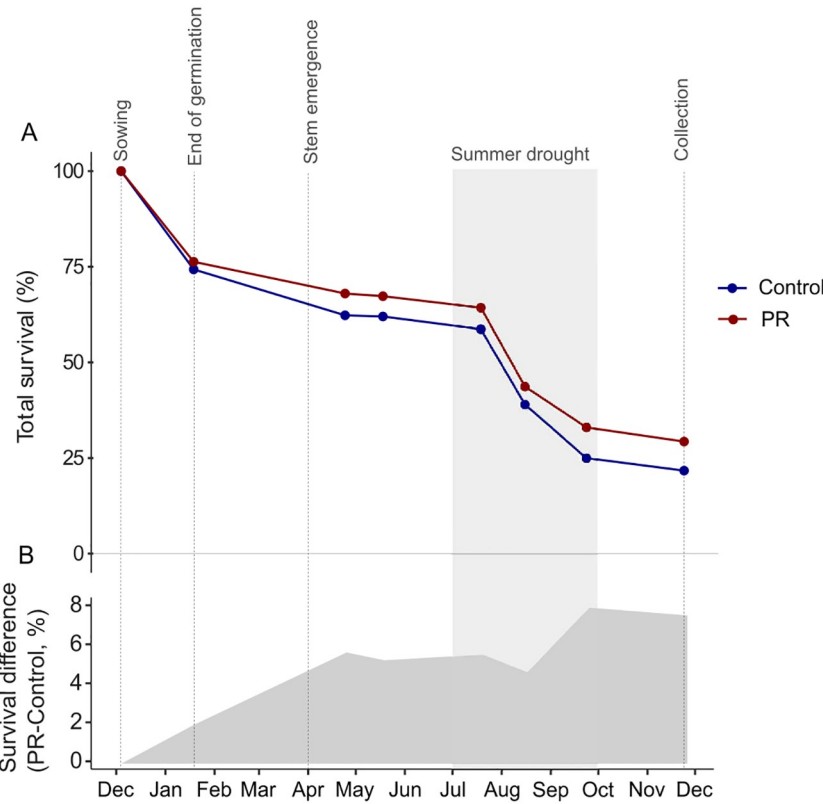

**Fig 3. Survival of seedlings per treatment during the experiment.** (A) Survival curves in the precipitation reduction treatment (PR, in red), and in the control (in blue). Values at "end of germination" corresponds to germination percentage in Fig 4A, and values at "Collection" corresponds to total survival (S_T) in Fig 4B. (B) Grey polygon represents the cumulative difference between the two survival curves, total survival in PR minus total survival in Control. The drought period of the year 2018 based on SWC data is highlighted in clear grey.

originating from heavy seeds and growing with relatively favourable light availability, they recruited less ECMsm.

When looking at the relationship between ECM colonization and seedling performance and growth of emerged plants, we found that the proportion of ECMsm colonized root tips was positively correlated with seedling survival and stem height, while the proportion of ECMc was negatively correlated with shoot biomass (Table 2).

## Discussion

In this study we combined a long term precipitation reduction experiment established for 15 years with an acorn germination experiment to get further insights into the factors controlling germination, seedling establishment, growth and the links with ectomycorrhizal colonization in holm oak, a tree species structuring Mediterranean landscapes at low elevation in the western part of the Mediterranean basin [8].

### 1) The impact of the precipitation reduction treatment on survival

Edaphic drought due to precipitation reduction is known to prevent or limit the success of *Q. ilex* seedling recruitment at two important stages. First, acorns being desiccation-sensitive

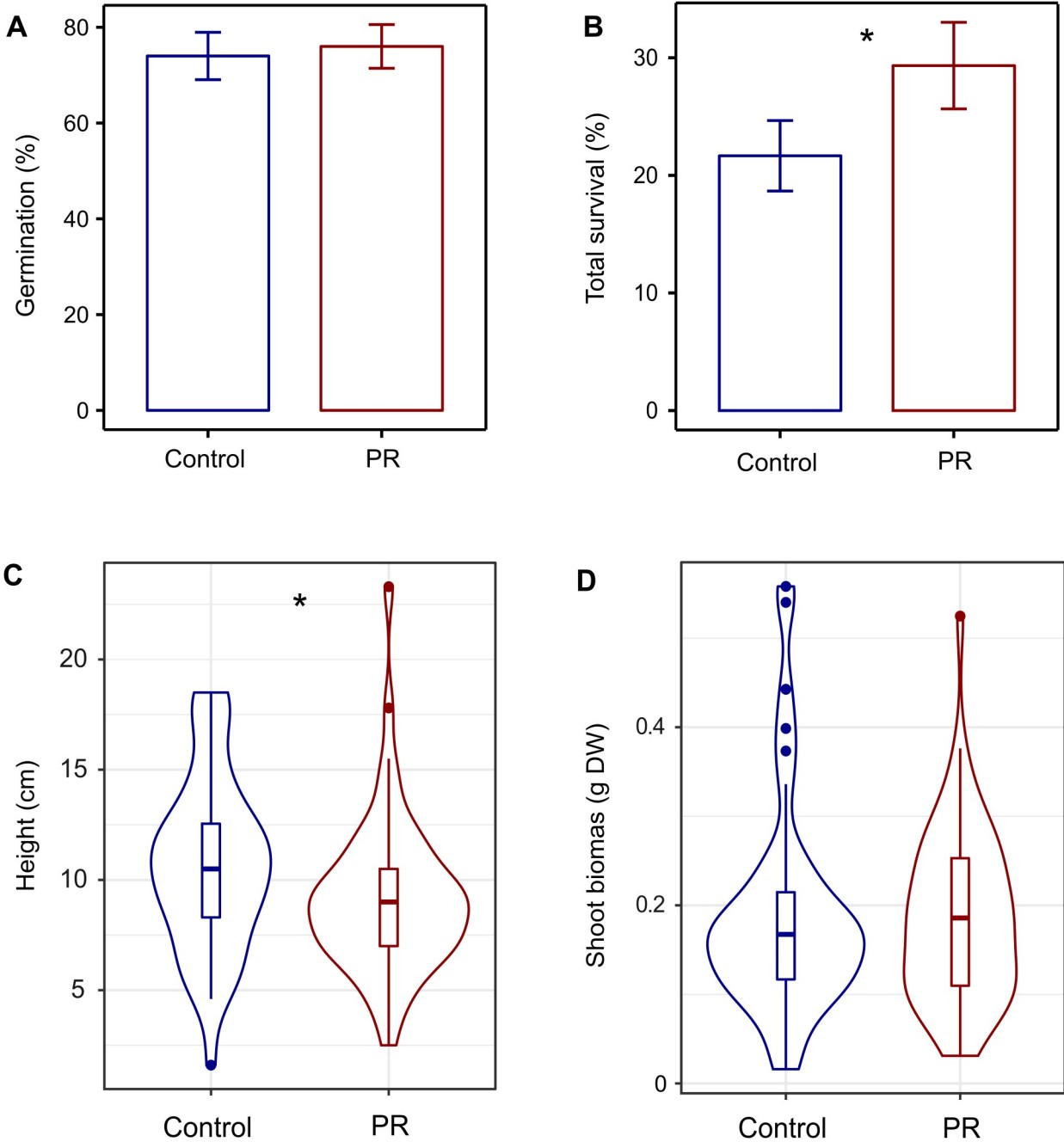

**Fig 4. Effect of precipitation reduction treatment on germination, survival, height and shoot biomass of seedlings.** (A) Germination and (B) Total survival ($S_T$), per cage n = 60. (C) Height measured at the end of the experiment (November 2018) per individual n = 153 (alive plants from all cages) and (D) shoot biomass per individual n = 96 (alive plants from half of the cages). *** $P<0.001$, ** $0.001<P<0.01$, * $0.01<P<0.05$, + $0.05<P<0.07$.

seeds, they may lose their ability to germinate if they dehydrate during winter dry spells [36]. Second, summer drought episodes that are characteristic of the Mediterranean climate, impose severe plant water stress that can be readily lethal to young seedlings [15]. In our experiment, the results showed that summer drought was the most important cause of mortality (Fig 3),

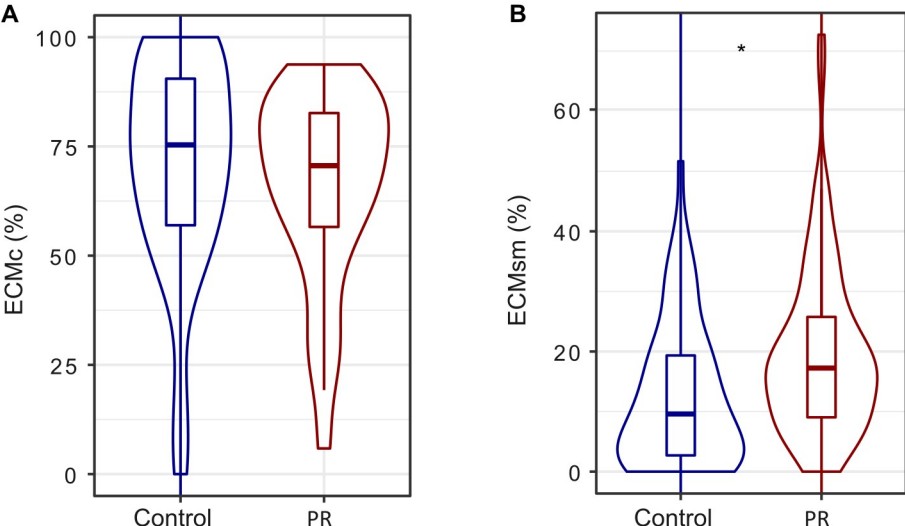

**Fig 5. Proportion of different ECM exploration types per total number of root tips.** (A) Contact ectomycorrhizal exploration type ECMc and (B) short and medium ectomycorrhizal exploration type ECMsm per individual n = 129 (all plants from half of the cages). *** P<0.001, ** 0.001<P<0.01, * 0.01<P<0.05, + 0.05<P<0.07.

thereby confirming the critical role of water stress in determining the success of seedling establishment. Failure to germinate was the second cause of mortality (Fig 3), being most likely not caused by winter seed dehydration since the winter of 2018 was relatively wet (49% of the mean annual precipitation was already fallen from January to March) and we kept the acorns in humid conditions before sowing in order to maximize their chances of germinating. However, in contrast to our first hypothesis, the total seedling survival ($S_T$) after 11 months was higher in the PR treatment than in the Control with ambient conditions (Fig 4B). Additionally, the survival of emerged plants ($S_{EP}$) was higher for seedlings growing in soil microhabitats

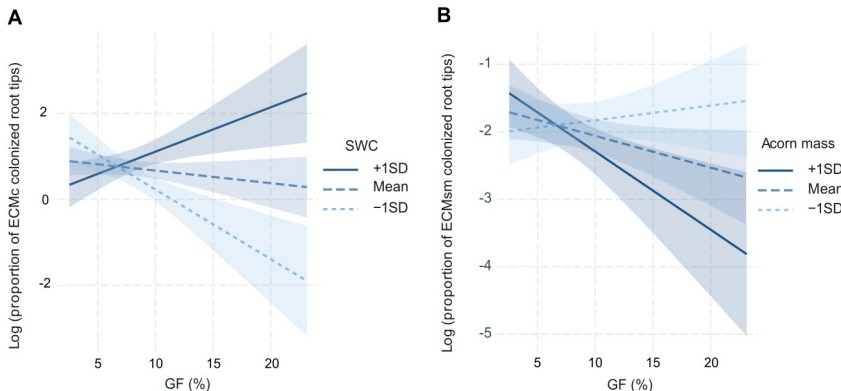

**Fig 6. Significant interactions from mixed linear models of ECM colonization (Table 1).** Canopy gap fraction (GF) interacts with soil water content (SWC) or acorn biomass to affect the proportion of mycorrhized root tips with (A) contact (ECMc) and (B) short and medium ectomycorrhizal (ECMsm) exploration types, per individual n = 129 (all plants from half of the cages). The three-way interactions present the relationship at three levels of the moderator variable (-1SD standard deviation, mean and +1SD standard deviation), which is SWC in (A) and acorn biomass in (B).

**Table 2. Model estimates and significance levels from linear mixed-effect models of ECM root tip colonization.**
The impact of contact ectomycorrhizal exploration type (ECMc) and short + medium ectomycorrhizal exploration type (ECMsm) on seedling survival, height and aerial biomass. $R^2m$ represents the variance explained by the fixed factors and $R^2c$ the variance explained by random and fix factors. *** $P<0.001$, ** $0.001<P<0.01$, * $0.01<P<0.05$, + $0.05<P<0.07$.

| Variable | ECMc | ECMsm | $R^2m$ | $R^2c$ |
|---|---|---|---|---|
| Survival emerged plants ($S_{EP}$) | 1.334 | 3.675* | 0.05 | 0.08 |
| Height survived | | 4.776* | 0.03 | 0.28 |
| Shoot biomass survived | -0.063* | | 0.03 | 0.13 |

with lower soil moisture. These two results indicate that, although summer drought was the main cause of mortality in both treatments, the partial rain exclusion treatment did not further increase seedling mortality. Instead, we found higher mortality in control plots during spring and at the end of the summer drought. Although surprising, negative effects of higher soil moisture conditions on seedling survival have been previously reported [37, 38], and were explained by sporadic water logging hampering radicle development and respiration. Water-logging does not readily occur at our study site because the rocky soil of the experimental site (that includes over 70% calcareous stones), allows for a very fast water infiltration into the deeper soil horizons even during very heavy rainfall events. Therefore, we argue that the higher seedling mortality under the wetter soil conditions of the Control treatment may have been rather related to biotic factors such as soil pathogens. In particular, root infection by Oomycetes is favoured by high soil moisture needed for sporulation and zoospore dispersion [39]. Plantations and natural forests of *Q. ilex* have indeed been severely damaged by the exotic and highly destructive Oomycete root pathogen *Phytophthora cinnamomi* [40, 41], which has been proposed as the main factor for oak decline in Spain and Portugal; and the high mortality of *Q. ilex* seedlings growing in *P. cinnamomi* infested soils has been observed after waterlogging combined with subsequent water deprivation [42]. Relatedly, important root damage was observed at higher levels of soil moisture in *Quercus suber* seedlings growing in soils inoculated with *P. cinnamomi* [43]. Therefore, one possible explanation that is in line with the mortality patterns observed in Fig 3 as well as the higher mortality in the Control is that the wetter soil conditions increased the likelihood of root infection with oomycetes. This presumably started during root system establishment in spring before the soil started drying (Fig 3). Thereafter, during the summer drought, a slightly higher soil moisture could have been even more detrimental as it might still allow the maintenance of oomycetes, thereby explaining the higher mortality found in the Control compared to PR. Furthermore, plants infected with oomycetes-damaged root tips could then also be more vulnerable to drought in late summer because they may be less colonized with drought-alleviating root mutualists such as ECM. This emerging conjecture needs, however, to be further tested with specific analyses of the pathogen presence and abundance in our experimental site as well as the presence of lesions on plant tissues.

## 2) Effects of microhabitat environmental conditions on survival and growth

Consistent with our second hypothesis, microhabitat conditions were found to be important drivers of early establishment success. Light and soil moisture conditions measured at the microhabitat (cage) level were retained as significant predictors in 7 out of the 8 response variables we assessed (Table 1). In contrast, the PR treatment was retained for only 3 out of 8 variables. This clearly indicates the importance of including the actual microhabitat conditions in studies aiming to understand germination and seedling survival. Our results show that light

affected positively the total survival of seedlings (from germination to collection of seedlings) but not the survival of the already emerged plants (from stem emergence to collection of seedlings). This indicates that the positive effect of light takes mainly place in the first stages of life (from germination to stem emergence and shoot development), but once stems emerged and have to deal with the summer drought the effect of higher light exposure can be detrimental as it can increase seedling water stress. These results agree partially with earlier studies [18] reporting that seedling germination and emergence increased linearly with light availability, but that seedling survival decreased because the light availability increases the risk of desiccation of seedlings in drier sites [15]. Pathogen infection was also presumably responsible for the negative effect of soil moisture on plant aerial biomass, since the infection could reduce growth.

Another finding of this study is that acorn biomass, here interpreted as the initial resources present in the acorns, is one of the most consistent predictors of germination, survival and growth. This is in line with several studies performed with *Quercus* sp. [44–46] and confirms earlier evidence that seed size is one of the most important traits influencing the early phases of the plant's life cycle. During the first year of their life, young *Q. ilex* seedlings have been shown to depend more markedly on their seed reserves than on the environmental conditions where they grow [47]. It is likely that once these reserves are consumed, microhabitat conditions will gain in importance during later stages of seedling growth and survival beyond the first year after germination.

## 3) Effects of mycorrhiza exploration types on survival

The PR treatment favoured the root colonization of young seedlings with ECM exploration types that have longer mycelia (short and medium exploration types, ECMsm) and decreased the colonization with contact ECM (ECMc, Fig 5). This is in line with our third hypothesis stating that seedlings growing in more water limited conditions develop typical drought-adapted ECM morphotypes with differentiated rhizomorphs because these have a better capacity to forage nutrients and water from further distances [26, 27, 48]. Apart from their function in increasing nutrient foraging, ECM can also act as biological deterrents to root pathogens as there is evidence that extended extramatrical mycelia and denser mycelia mantles can also act as physical barriers to pathogen infection. In fact, [49] found lower abundance of extramatrical hyphae and less dense mantles in *Phytopththora cambivora* infected and declining chestnut stands, compared to non-infected chestnut stands. Similarly, [50] found that declining *Quercus robur* stands were associated with less long-distance and more short-distance exploration types (contact, short and medium) than what they found in healthier *Q. robur* stands. This potential role of ECM species with longer mycelia in reducing pathogen infection could help to understand the positive relation between ECMsm and seedling survival and height found in our experiment (Table 2). Furthermore, some ECM species have the ability to decrease the virulence of *Phytophthora* through exudation of antibiotics or terpenes with deleterious effects on pathogens [51]. Several studies comparing the declining status of *Q. ilex* stands revealed a negative correlation between ECM abundances and the abundances of pathogens and saprobes [52]. In the same vein, a decrease in vital root tips (ectomycorrhized with well-developed mantles) combined with an increase of non-vital root tips (non-mycorrhized and deteriorated roots) that are more vulnerable to pathogen infection was found in declined *Q. ilex* stands [53]. At our study site, significant changes in ECM community composition, but not in total species richness, were found in the PR treatment compared to the Control [25]. It could be hypothesized that the ectomycorrhizal community in the PR treatment is the result of the selective pressure exerted by the long-term reduced water availability that favoured ECM

species with abundant/more extramatrical mycelium, which may improve seedling persistence under drought conditions even if they require more carbon investment by the host seedlings. These drought-adapted ECM species may increase seedling survival in PR treatment through higher forage capacity, but also by being physical barriers against root pathogens. However, such potential mechanisms of how different ECM species may affect seedling performance needs more detailed studies, and our data does not allow to infer any causal relationships. The same for the negative correlation between shoot biomass and contact ECM exploration type (Table 2), the biological mechanisms involved are not clear and should be investigated in controlled experiments testing the composition and richness of contact morphotypes affecting growth.

In terms of micro-environmental conditions, light availability was found to modulate ECM colonization in interaction with soil moisture or initial seed mass depending on the type of mycorrhiza exploration (Fig 6). Light had a negative effect on ECMc in drier microsites, but a positive effect in more humid microsites. This is consistent with the hypothesis that light modulates the soil drying through the evaporative demand, so that being under shadow in drier soils could be beneficial to maintain soil humidity and thereby increase the chances of being colonised by contact mycorrhiza that are more adapted to more humid soils. Besides, the interactive effect of seed mass and light on ECMsm suggests that even root colonization by mycorrhizae could be affected by the initial acorn resources, which is an aspect that, to the best of our knowledge, has not been reported previously.

## Conclusions

Although summer drought was the main cause of seedling mortality, our study indicates that drier conditions in spring can increase seedling survival, presumably through a synergistic effect of drought adapted ECM species and less favourable conditions for root pathogens infection. However, it is important to put this finding in context as the particularly wet spring and autumn periods in 2018 (with frequent rain events) combined with a typical severe Mediterranean drought period in summer, all of which represent favourable conditions for pathogen infection in *Q. ilex* [42]. Notably, acorn biomass was one of the most consistent variables positively affecting germination, survival and growth, and it even influenced ECM colonization. The regeneration of *Q. ilex* stands could thus be threatened if the more intense and longer summer droughts expected in the near future, reduce the acorn biomass by increasing the water stress experienced by acorn bearing trees [54–56].

## Supporting information

**S1 Fig. The effect of SWC (% volumetric) per cage in survival of emerged plants per individual.** n = 391.
(TIF)

**S1 Rawdata.**
(XLSX)

## Acknowledgments

The authors gratefully acknowledge Patrick Schevin, Quentin Lassus, and Adrien Millan for helping with the field work. The authors are also indebted to Jean-Marc Ourcival who designed and maintained the long-term rainfall exclusion experiment in Puéchabon throughout the years, and to David Degueldre, Thierry Matthieu, Pauline Durbin and Karim Piquemal for their help. The Puéchabon experimental site belongs to the OSU OREME (UMS 3282) and

is annually supported by the research infrastructure AnaEE-France (ANR-11-INBS-0001) and by Allenvi through the SOERE F-ORE-T.

## Author Contributions

**Conceptualization:** Laura García de Jalón, Jean-Marc Limousin, Franck Richard, Arthur Gessler, Martina Peter, Stephan Hättenschwiler, Alexandru Milcu.

**Data curation:** Laura García de Jalón, Jean-Marc Limousin, Franck Richard, Alexandru Milcu.

**Formal analysis:** Laura García de Jalón, Alexandru Milcu.

**Funding acquisition:** Alexandru Milcu.

**Investigation:** Laura García de Jalón, Jean-Marc Limousin, Franck Richard, Alexandru Milcu.

**Methodology:** Laura García de Jalón, Jean-Marc Limousin, Franck Richard, Alexandru Milcu.

**Supervision:** Jean-Marc Limousin, Alexandru Milcu.

**Writing – original draft:** Laura García de Jalón.

**Writing – review & editing:** Laura García de Jalón, Jean-Marc Limousin, Franck Richard, Arthur Gessler, Martina Peter, Stephan Hättenschwiler, Alexandru Milcu.

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
