## [Decision Letter · Decision Letter 0]

26 Mar 2020

PONE-D-20-04357

Microhabitat and ectomycorrhizal effects on the establishment, growth and survival of Quercus ilex L. seedlings under drought

PLOS ONE

Dear Garcia de Jalon,

Thank you for submitting your manuscript to PLOS ONE. After careful consideration, we feel that it has merit but does not fully meet PLOS ONE’s publication criteria as it currently stands. Therefore, we invite you to submit a revised version of the manuscript that addresses the points raised during the review process.

We would appreciate receiving your revised manuscript by May 10 2020 11:59PM. To enhance the reproducibility of your results, we recommend that if applicable you deposit your laboratory protocols in protocols.io, where a protocol can be assigned its own identifier (DOI) such that it can be cited independently in the future. For instructions see: http://journals.plos.org/plosone/s/submission-guidelines#loc-laboratory-protocols

We look forward to receiving your revised manuscript.

Kind regards,

Ines Ibáñez, Ph.D.

Academic Editor

PLOS ONE

Journal Requirements:

Additional Editor Comments (if provided):

This is a sound study of interest within this area of work. However, before publication, the authors need to clarify and justify their methodology, i.e., the use of dead seedlings, and to clarify their hypothesis and focus of their analysis.

Reviewers' comments:

Reviewer's Responses to Questions

**Comments to the Author**

1. Is the manuscript technically sound, and do the data support the conclusions?

Reviewer #1: Partly

2. Has the statistical analysis been performed appropriately and rigorously? 

Reviewer #1: Yes

3. Have the authors made all data underlying the findings in their manuscript fully available?

Reviewer #1: Yes

4. Is the manuscript presented in an intelligible fashion and written in standard English?

Reviewer #1: Yes

5. Review Comments to the Author

Reviewer #1: Summary

This paper examines the impact of drought, microhabitat, and ectomycorrhizae (ECM) on the germination, survival, and growth of an evergreen oak seedling. To do so the authors compared seedlings grown in a long-term precipitation reduction treatment, measuring soil water content and light availability at the microhabitat scale. The authors also examined the ECM colonization of seedlings, noting the proportion of different morphotypes. The authors found that total survival was higher under the precipitation exclusion experiment, and that seedlings in drier soils were colonized by ECM with hyphae that extended farther from the seedlings’ root tips.

Major Issues

I was not totally sure what the authors were hypothesizing with their second hypothesis. I believe I understood the basis of the hypothesis - that microhabitat environmental factors are more important in determining seedling survival – however the phrasing of it made it unclear. The authors use the term mechanism in this hypothesis but only once more throughout the paper, I think the use of this word without introducing it earlier in the introduction is what through me as a reader off.

I am curious about the use of both alive and dead seedlings for the ECM analysis. In other literature I have only seen seedlings alive at harvest used. Is it possible that using dead seedlings in this part of the analysis could alter the results, as a different suite of fungi may colonize dead seedlings? Could this be an explanation of the results that you found here? Additional information on this, including data about whether or not the ECM community differed between alive and dead seedlings would help alleviate this concern.

In the results section when the authors discuss the effects of soil water content and light availability it is unclear whether these results are based upon means at the treatment level (i.e. precipitation reduction vs. control) or at the seedling level (i.e. microhabitat). As this is a key point that the paper is attempting to make, clarifying this is important in order to draw the conclusion the authors are drawing.

Minor Issues

Overall, the paper is well written, however there were a few typos and grammatical errors throughout the paper that need to be addressed. Some clarification on figures is also necessary. I’ve listed all minor issues below:

• Line 52:54 – confusing sentence

• Line 57:59 – citation?

• Line 59:61 – citation?

• Line 63:65 – “with different soil moistures” strikes me as an odd way to put what they are trying to say; could overall use a better description of the two cited studies

• Line 91:94 – which exploration type is this?

• Line 120 – typo, understorey

• Line 204 – Could it be the harvesting techniques caused the lack of long-distance exploration type?

• Figure 1

o Are differences significant across every measurement point?

o Where is data for June and July?

• Figure 2

o Are differences significant for both April and September as well as overall?

o Should X axis label of “Day of Year” say month?

• GF results (line 245:249 and Figure 2) – only measured in April and September, you mention the canopy is dominated by evergreen oaks, but are there other deciduous plants that could be impacting light availability between these two times?

• Table 1

o Line 259 – typo “It is tested…”

o Effect of GF on germination, there is no symbol for significance, is this intentional?

• Line 270 – typo “it was significant lower…”

• How are seedling mortality during germination and germination rate different?

• Line 273:277 – are these differences statistically significant?

• Figure 3

o You mention drought conditions, was the summer drier than a typical summer, or is this just the dry period typical of a Mediterranean climate?

• Why did you not measure belowground biomass?

• Figure 6

o Line 319 – typo “it is showed…”

o Overall, I found this figure to be very difficult to understand, does +1/-1 SD mean you are only using those values in that range?

• Table 2

o Line 327:328 – typo “it is tested…”

• Line 338 – What basin are you referencing here?

• Line 380 – can you see oomycete damage under the dissecting scope? If you can did you notice any damaged root tips while counting ECM colonization?

• Line 399:42 – this doesn’t necessarily agree with what the results from this paper show, you found no impact of light on germination survival, and a positive impact on total survival, and no impact on emerged survival

• Line 402:403 – could lower biomass in seedlings with more SWC also be due to the different morphotypes of ECM colonizing them?

• Line 467:471 – this conclusion seems to come a little out of nowhere as you have not discussed any of this previously, why bring it up now at the very end of the paper?

6. PLOS authors have the option to publish the peer review history of their article (what does this mean?). If published, this will include your full peer review and any attached files.

Reviewer #1: No

---

## [Author Response · Author response to Decision Letter 0]

15 Apr 2020

We would like to thank the editor and the reviewer for the time invested in this manuscript and the constructive criticism. We took into account all the suggestions and comments, and we present a thoroughly revised manuscript with specific focus on clarifying the methodology and the hypotheses. Furthermore, as requested by the reviewer, we present additional analyses.

A detailed point-by -point reply addressing the reviewer comments could be found in the "Response to Reviewers" file.

---

## [Decision Letter · Decision Letter 1]

19 May 2020

Microhabitat and ectomycorrhizal effects on the establishment, growth and survival of Quercus ilex L. seedlings under drought

PONE-D-20-04357R1

Dear Dr. Garcia de Jalon,

We are pleased to inform you that your manuscript has been judged scientifically suitable for publication and will be formally accepted for publication once it complies with all outstanding technical requirements.

With kind regards,

Ines Ibáñez, Ph.D.

Academic Editor

PLOS ONE

Additional Editor Comments (optional):

The authors have successfully addressed the issues brought by the review, now there are only minor changes needed.

Reviewers' comments:

Reviewer's Responses to Questions

**Comments to the Author**

1. If the authors have adequately addressed your comments raised in a previous round of review and you feel that this manuscript is now acceptable for publication, you may indicate that here to bypass the “Comments to the Author” section, enter your conflict of interest statement in the “Confidential to Editor” section, and submit your "Accept" recommendation.

Reviewer #1: All comments have been addressed

2. Is the manuscript technically sound, and do the data support the conclusions?

Reviewer #1: Yes

3. Has the statistical analysis been performed appropriately and rigorously? 

Reviewer #1: Yes

4. Have the authors made all data underlying the findings in their manuscript fully available?

Reviewer #1: Yes

5. Is the manuscript presented in an intelligible fashion and written in standard English?

Reviewer #1: Yes

6. Review Comments to the Author

Reviewer #1: Excellent work, really appreciate the time taken to address all of my previous comments. You're answers were completely satisfactory, and alleviated the concerns I had.

My only additional note is that in the discussion lines 435 to ~450 you are inconsistent with the use of mantel and mantle when discussing ECM physiology.

7. PLOS authors have the option to publish the peer review history of their article (what does this mean?). If published, this will include your full peer review and any attached files.

Reviewer #1: No

---

## [Editor Report · Acceptance letter]

28 May 2020

PONE-D-20-04357R1 

Microhabitat and ectomycorrhizal effects on the establishment, growth and survival of Quercus ilex L. seedlings under drought 

Dear Dr. Garcia de Jalon:

I am pleased to inform you that your manuscript has been deemed suitable for publication in PLOS ONE. Congratulations! Your manuscript is now with our production department. 

With kind regards,

on behalf of

Dr. Ines Ibáñez 

Academic Editor

PLOS ONE